# New Insights into Hemopexin-Binding to Hemin and Hemoglobin

**DOI:** 10.3390/ijms23073789

**Published:** 2022-03-30

**Authors:** Guilherme C. Lechuga, Paloma Napoleão-Pêgo, Carlos M. Morel, David W. Provance, Salvatore G. De-Simone

**Affiliations:** 1Center for Technological Development in Health (CDTS), National Institute of Science and Technology for Innovation on Neglected Population Diseases (INCT-IDPN), FIOCRUZ, Rio de Janeiro 21040-900, RJ, Brazil; gclechuga@gmail.com (G.C.L.); pegopn@gmail.com (P.N.-P.); carlos.morel@fiocruz.br (C.M.M.); bill.provance@fiocruz.br (D.W.P.); 2Laboratory of Epidemiology and Molecular Systematics (LESM), Oswaldo Cruz Institute, FIOCRUZ, Rio de Janeiro 21040-900, RJ, Brazil; 3Department of Cellular and Molecular Biology, Biology Institute, Federal Fluminense University, Niterói 24020-141, RJ, Brazil

**Keywords:** hemopexin, hemoglobin, protein-protein binding, hemin

## Abstract

Hemopexin (Hx) is a plasma glycoprotein that scavenges heme (Fe(III) protoporphyrin IX). Hx has important implications in hemolytic disorders and hemorrhagic conditions because releasing hemoglobin increases the labile heme, which is potentially toxic, thus producing oxidative stress. Therefore, Hx has been considered for therapeutic use and diagnostics. In this work, we analyzed and mapped the interaction sequences of Hx with hemin and hemoglobin. The spot-synthesis technique was used to map human hemopexin (P02790) binding to hemin and human hemoglobin. A library of 15 amino acid peptides with a 10-amino acid overlap was designed to represent the entire coding region (aa 1-462) of hemopexin and synthesized onto cellulose membranes. An in silico approach was taken to analyze the amino acid frequency in the identified interaction regions, and molecular docking was applied to assess the protein-protein interaction. Seven linear peptide sequences in Hx were identified to bind hemin (H1–H7), and five were described for Hb (Hb1–Hb5) interaction, with just two sequences shared between hemin and Hb. The amino acid composition of the identified sequences demonstrated that histidine residues are relevant for heme binding. H105, H293, H373, H400, H429, and H462 were distributed in the H1–H7 peptide sequences, but other residues may also play an important role. Molecular docking analysis demonstrated Hx’s association with the β-chain of Hb, with several hotspot amino acids that coordinated the interaction. This study provides new insights into Hx-hemin binding motifs and protein-protein interactions with Hb. The identified binding sequences and specific peptides can be used for therapeutic purposes and diagnostics as hemopexin is under investigation to treat different diseases and there is an urgent need for diagnostics using labile heme when monitoring hemolysis.

## 1. Introduction

Hemopexin is a plasma glycoprotein that scavenges heme (Fe(III) protoporphyrin IX) with a high affinity [1]. It is a pivotal protein in hemolytic conditions and cell injury in malaria, sickle cell disease (SCD), thalassemia, hemorrhage, and hemorrhagic stroke [2]. Hemoglobin (Hb), a hetero-tetrameric protein, is the most abundant heme protein in the blood. When heme is released from hemoglobin, it binds to Hx or albumin with the lowest affinity. The detrimental effect of releasing hemoglobin (Hb) during hemolysis is prevented by the scavenger serum glycoprotein haptoglobin (Hp), a tetrameric protein with two α/β dimers that scavenge free Hb with high avidity [3]. The complex Hp-Hb is subsequently degraded in hepatocytes and macrophages by binding to a specific receptor CD163 [4]. Unbound hemoglobin undergoes oxidation and can release heme that contains iron. When heme is not associated with serum proteins, it potentially produces oxidative stress through the Fenton reaction [5]. Therefore, releasing high amounts of hemoglobin leads to exhaustion of the haptoglobin (Hp)-binding capacity and increases the heme concentration that saturates Hx. The unbound free heme induces oxidative damage that produces severe complications, including stroke, thrombosis, and pulmonary embolism [5,6,7]. Increasing evidence demonstrates the importance of Hx, and knockout mice showed neuronal degeneration and cognitive impairment after induction of intracerebral hemorrhage [8]. Hx also prevents heme-induced proinflammatory switching of macrophages toward an M1 phenotype [9]. Recently, heme and iron dysmetabolism was observed in Alzheimer’s disease (AD), with a significant increase of hemoglobin subunit β in AD [10].

Crystallographic studies showing the three-dimensional structure of the rabbit hemopexin-heme complex and revealed that Hx has four-bladed β-propeller folding motifs that are similar in the N- and C-terminal domains, joined by a flexibly structured hinge sequence. Furthermore, heme has binding sites in histidine residues along the flexible linker region and another in the C-terminal domain [11]. However, interactions can occur with other amino acids. Additionally, essential functions of the complex heme-Hx have also been highlighted, like serine protease and immunomodulatory activity [4]. 

An open question is how Hx could interact with hemoglobin, the primary source of heme in blood. Studies highlight that heme moiety can be transferred from hemoglobin to Hx. This transfer preferentially occurs when heme is oxidized to its ferric form (methemoglobin). Although heme extraction of Hb by Hx has been reported, the exact mechanism of heme transfer at the molecular level remains unclear [12,13].

Considering its significant biological implications, it is crucial to understand the interactions of Hx with heme and hemoglobin. Furthermore, identifying the Hx interactome may substantially impact medicine since Hx has been reported as a biomarker for hemolysis and sepsis [14]. Administration of Hx can also improve the clinical outcome of some diseases, e.g., sickle cell disease [15]. Accordingly, the utilization of recombinant Hx, plasmapheresis, or heme-binding peptides has commercial potential for the treatment of the above-cited pathologies. Here, we applied a microarray of peptides and in silico analysis to map the interactions of Hx with hemin and hemoglobin; seven linear peptide sequences in Hx were identified to bind hemin, and five were depicted for the Hb interaction. 

## 2. Results

### 2.1. Identification of the Hemopexin-Hemin and Hemopexin-Hemoglobin Interaction Regions

To evaluate the interaction regions of Hx with hemin and Hb, we employed the strategy of parallel synthesis of peptides on a cellulose membrane. Spot-synthesis analysis was used with an array of 15-mer peptides with a 10-amino acid overlap, representing the entire Hx sequence synthesized in situ on a cellulose membrane. Following incubation with hemin/Hb and its subsequent detection by the hemin peroxidase-like activity and Hb-specific antibodies with corresponding secondary antibodies, several highly reactive spots were revealed that indicated a direct interaction between sequences in Hx with hemin and Hb (Figure 1). From the calculated relative intensity percentage, where signal intensities above 50% were considered the cutoff for a positive reaction, seven individual peptide sequences were defined that interacted with hemin (Figure 1A) and five with Hb (Figure 1B). Secondary structural analysis revealed that most of the interaction sequences in Hx were coiled in their arrangement (7), while a few were also in beta-sheets (3). Just one sequence that interacts with Hb (Hb1) was in the alpha-helix structure located in the N-terminal domain (Table 1), but most of the sequences (Hb2–Hb5) were in the coil. Sharing only two Hx interaction sequences between hemin and Hb (Figure 2), peptide fragments **RLHIMAGRRL** and **KSGAQATWTE** were present in H4 and H5, respectively, and Hb5. 

### 2.2. Frequency of Amino Acids in Hemopexin-Hemin/Hemoglobin Binding Sites

Interaction with heme occurs mainly with histidine residues. The analysis of the amino acid frequency in all Hx peptide sequences showed that leucine (L; 15.2%) was the most frequent amino acid in hemoglobin interaction sites, followed by alanine (A, 10.5%), tryptophan (W, 8.6%), and glycine (G, 8.6%). Glycine (G, 10.5%) was also a frequent amino acid at Hx/hemin interaction sites, followed by leucine (L, 9.5%), proline (P, 9.5%), serine (S, 9.5%), and histidine (H, 8.4%; Figure 3A). Glycine, histidine, and serine were present in all Hx interaction sequences with hemin (H1–H7). In contrast, alanine, glycine, leucine, and tryptophan comprised 100% of Hx/Hb binding sequences (Hb1–Hb5; Figure 3B). Histidine residue is important in heme-binding proteins. H105, H293, H373, H400, H429, and H462 were distributed in the H1–H7 peptide sequences (Figure 2 and Table 1). 

### 2.3. Molecular Docking Analysis of Hemopexin and Hemoglobin

This interaction was also inferred from a molecular docking analysis for protein-protein interactions. The interactome of Hx was evaluated using network-based analysis with selected highest-confidence interactions. The results pointed out that Hx can interact with others serum proteins like albumin (ALB), haptoglobin (HP), and the β-chain of hemoglobin (HBB). Additionally, human Hx can interact with the fibrinogen gamma chain (FGG), metalloproteinase inhibitors 1 and 2 (TIMP1 and TIMP2), low-density lipoprotein receptor-related protein 1 (LRP1), and feline leukemia virus subgroup C receptor-related protein 1 (FLVCR1), a receptor-related to heme export. A molecular docking assay evidenced the possibility of binding the Hb beta domains. The molecular model indicated a close association between the Hx and the β-chain of Hb (Figure 4B,C). Interaction regions showed that five peptide sequences bind strongly to human hemoglobin chains. However, Hb5 (RLHIMAGRRLWWLDLKSGAQATWTE) had the highest reactivity signal in spot-synthesis, suggesting that this region could contribute to Hb-Hx binding. Hb5 contains two peptide sequences (H4 and H5) that bind strongly to hemin. Since this region shares both hemoglobin and hemin interaction, we hypothesized that these residues could mediate the contact region with Hb and heme.

An in silico information-driven flexible docking approach was used to study Hx-Hb interactions, specifically the transfer of heme to Hx. Several interaction poses were clustered, creating 126 structures in 13 clusters, which represented 63% of the water-refined models generated (Appendix A). According to the HADDOCK web server, cluster 8 was considered the most reliable for rank structures by biochemical and/or biophysical data (Appendix A). The structures in cluster 8 were carefully analyzed (Appendix A), and the lowest-energy structure (−102 Kcal/mol) was chosen for further analysis. Hotspot analysis showed that this interaction could be coordinated by six amino acid residues from Hx (SER370, ARG371, TRP382, ASP384, THR392, and THR394) and six from Hb (LYS65, LYS66, ASP73, THR84, THR87, and THR88; Figure 4C and Appendix A). In addition, hydrogen bonds were evidenced at the protein-protein interface between Hx and the Hb β-chain (Table 2). Several residues in Hx interacted with propionate heme groups, forming hydrogen bonds (SER387, ASP384, and THR392; Figure 4D). Hydrophobic interactions were evidenced (Appendix A). 

## 3. Discussion

Understanding the hemopexin binding motifs with heme and the interactions with other proteins are crucial for determining its function, implications in physiology and pathology, and potentially opening up new avenues for disease treatment and diagnostics. The binding of mammalian hemopexin to heme was studied by X-ray crystallography, and the 3D structure of the rabbit hemopexin-heme complex revealed similar N and C-terminal domains, with four-bladed β-propeller domains that are united by a hinge sequence [16]. Heme was shown to bind to Hx by coordination of His residues (H236 and H293) in the linker region [16]. Interestingly, H293 was present in the H3 (GWHSWPIAHQ) peptide sequence with a high hemin binding capacity (signal of ~50%), and H236 also interacted with hemin in the B22–B24 sequences (DYFMPCPGRGHGHRN, CPGRGHGHRNGTGHG, and HGHRNGTGHGNSTHH), but with a lower affinity, with the signal ranging from 27.5 to 37%. Additionally, Hx-heme interactions were revisited by a computational and experimental study that showed histidine residue sites that can bind heme [11]. Similarly, our data demonstrated that hemin binds with residues in Hx binding pocket H236/H293. All peptide sequences that interacted with hemin had histidine residues; notably, H373 was present in the peptides with the highest signal spots (>90%). Works suggest binding heme axially between H105 and H150 [10,13], but our data demonstrated that only H105 interacted with hemin in H2 (FRQGHNSVFL). Many molecular interactions can mediate heme coordination, which can occur by hydrogen bonds via propionate heme groups, π-π stacking between rings, and electrostatic and hydrophobic interactions. The most relevant heme-coordinating amino acids are histidine, cysteine, tyrosine, methionine, and lysine at a lower frequency [17,18]. 

The identified Hx peptide sequence that binds hemin can potentially be used for diagnostics. There is a gap in the reliable diagnostic tests for determining the levels of free heme in biological samples. The severity of hemolysis and heme toxicity may currently be indirectly estimated by the Hp and Hx levels [14,19]. Thus, a practical and specific test is urgently needed to determine the labile heme in biological fluids and tissues. Hx identified peptides sequences that could be used in platforms such as nanoparticles [20] for heme-scavenging therapy, to improve the clinical outcomes for hemolytic diseases like malaria, sickle cell disease (SCD), thalassemia, and hemorrhagic conditions. This vectorization could direct these specific heme-binding peptides to cellular compartments. The scavenging of labile heme by Hx blocked heme-driven tumor growth and metastases in a model of prostate cancer [21]. Recently, the association of Hb and heme levels with AD [10] has been proposed to bring about new applications for heme-binding protein, to prevent this neurodegenerative disease. In addition, Hb and heme were implicated in a murine model of severe sepsis [22]. Another application of hemopexin is to prevent acute adverse effects in blood transfusions after prolonged storage of red blood cells (RBC); transfusion of prolonged packed RBC can increase morbidity and mortality in trauma-induced hemorrhage, mainly by alterations to the RBC and hemolysis [23]. Hx can be administered intravenously but has potential side effects caused by its structure and protease activity [24,25,26]. Thus, it is crucial to map only heme-binding sites in the Hx structure, to abolish potential risky side effects. 

Hemoglobin is the primary source of heme in our blood. When investigating the possible interaction between Hx and Hb, analysis of the interactome database showed binding to the β-chain of Hb and other plasmatic proteins such as albumin and haptoglobin. Previously, data demonstrated that Hx removes heme from Hb, generating protein aggregates [27]. Heme group transfer to hemopexin is coordinated preferentially when heme moieties are oxidized to their ferric form, and it appears to proceed in a four-stage process until hemopexin successively transfers all the four heme groups from the hemoglobin tetramer, first binding to the Hb β chains, followed by loss from the Hb α chains [12,13]. Heme can be extracted from Hb by pathogens. *Staphylococcus aureus* has nine iron-regulated surface determinants (Isds), proteins that directly extract heme from hemoglobin (Hb) [28]. These proteins are structurally related and contain a variable number of conserved-near-iron transporter (NEAT) domains. These domains are distinct; some motifs are involved in Hb binding, and others bind heme that contains a conserved serine and YXXXY motif [29]. A study showed that a purified NEAT domain fused with a human haptoglobin β-chain extracted heme from hemoglobin and reduced the hemoglobin content and peroxidase activity [30]. Molecular dynamic simulations revealed that Y646 and Y642 tyrosine residues coordinate with heme iron, and during the heme transfer, two serine residues (S557 and S563) capture heme [30]. Docking analysis showed a lack of the YXXXY motif at Hx interaction sites, but SER387 was observed to form hydrogen bonds with the propionate heme group, thus possibly representing an initial contact point for heme extraction. 

There are scarce data on protein-protein interactions (PPI) between Hb and Hx. Addition of a Hx to ferrihemoglobin is known to cause a shift in chemical equilibrium, with a resulting displacement of heme groups from hemoglobin [13]. Hemopexin rapidly binds with a very high affinity when free heme is available, but the free heme concentration is shallow and highly regulated [31]. Thus, unbound Hx could interact with hemoglobin. In hemolytic conditions, the hemoglobin concentration increases rapidly, followed by its degradation into heme. Hx may play a role in the initial scavenging of heme from Hb, providing a defense against the toxicity of extracellular hemoglobin. However, Hx extraction of heme from Hb potentiates Hb neurotoxicity by a pro-oxidant iron-dependent mechanism that precipitates globin chains when Hp is absent [27]. Additionally, previous work showed the immunomodulatory effect of the heme sequestration mechanism of hemopexin. It suppressed the synergistic inflammation effect of HMGB1, a nuclear and cytosolic DNA-binding protein released from damaged cells with hemoglobin and impaired cytokine release [32]. 

## 4. Materials and Methods

### 4.1. Spot-Synthesis

Uniprot retrieved the sequence of hemopexin (P02790). A library of 15 amino acid peptides with a 10-amino acid overlap was designed to represent the entire coding region (aa 1-462) of hemopexin and synthesized onto cellulose membranes using an Auto-Spot Robot ASP222 (Intavis, Koeln, Germany) [33,34]. The synthetic peptide membranes were washed with TBS-T and then blocked with TBS-T containing 1.5% BSA under agitation for 2 h at room temperature. After extensive washing with TBS-T (Tris-buffer saline, 0.1% Tween 20, pH 7.0), membranes were incubated overnight with Hb (5 µg/mL) dissolved in TBST + BSA (0.75%) or hemin (10 µM) for 2 h. After incubation, membranes incubated with hemin were washed with TBS three times for 5 min, and then hemin-peroxidase-like activity was revealed using chemiluminescent substrate Super Signal R West Pico. Membranes incubated with hemoglobin were washed with TBS-T, followed by additional incubation with anti-human Hb antibody for 90 min. Subsequently, the membrane was washed with TBS-T and incubated for 90 min with an anti-rabbit IgG antibody conjugated to alkaline phosphatase (Sigma-Aldrich, Saint Louis, MO, USA), diluted 1:5000 in TBS-T solution containing 0.75% BSA. Washes were performed with TBS-T and then the substrate for chemiluminescent alkaline phosphatase Tropix^®^ was added. Next, the membrane was washed three times with TBS-T, and then the buffer was exchanged with CBS (50 mM citrate-buffer saline) before adding the chemiluminescent enhancer Nitro-Block II. Chemiluminescent signals were detected on an Odyssey FC (LI-COR Bioscience, Lincoln, NE, USA) using the same conditions described previously [33] with minor modifications. Briefly, a digital image file was generated at a resolution of 5 MP, and the signal intensities were quantified using TotalLab TL100 (v 2009, Nonlinear Dynamics, Newcastle-Upon-Tyne, UK) software [33]. Spots with a signal intensity greater than or equal to 50% of the highest signal value were considered to identify possible binding motifs.

### 4.2. In Silico Analysis

The peptides were uploaded to a webserver (https://web.expasy.org/protparam/; accessed on 21 February 2022). Analysis and graphical representation were carried out using R software version 4.1.1 and RStudio. The hemopexin interaction network was charted on the platform STRING (https://string-db.org/; accessed on 21 February 2022). The network nodes are proteins, and the edges represent the predicted functional associations of high confidence interactions. Protein 3D structures were visualized and analyzed using the PyMOL Molecular Graphics System, Version 2.0, Schrödinger, LLC. The hydron bonds and hydrophobic interactions between the two protein complexes were analyzed using Ligplot+ [35].

### 4.3. Molecular Docking

This assay was performed with a human hemopexin model, using Swiss-model (https://swissmodel.expasy.org/; PDB code—1qjs, accessed on 10 February 2022). Protein-protein interaction was assessed using the HADDOCK webserver (https://wenmr.science.uu.nl/haddock2.4/, accessed on 15 March 2022) with human hemopexin and the hemoglobin beta chain (PDB: 2m6z). HADDOCK2.4 is a platform for protein-protein interactions analysis. HADDOCK is a data-driven docking program guided by predicting likely residues (called ambiguous interaction restraints (AIRs)) found at the interface. These residues may be active (interacting residues) or passive (nearby interacting residues). Hemopexin’s active residues were defined as Hb5 (RLHIMAGRRLWWLDLKSGAQATWTE), retrieved from spot-synthesis data. The heme group residues in hemoglobin were selected as passive [36]. Protein-protein interaction was evaluated using the database (https://mitchellweb.ornl.gov/KFC_Server/, accessed on 20 February 2022) for hotspot amino acid residues’ prediction. 

## 5. Conclusions

These new insights may contribute valuable information on Hx’s structure, motifs, and protein-protein interactions with Hb. Binding sequences can be applied for therapeutic purposes and diagnostics. Further opportunities for research include the synthesis of the identified peptides for binding affinity measurement and overall molecular analysis to depict, using molecular dynamic simulations, the mechanism of heme transfer to Hx. Hemopexin is under investigation to treat different diseases, mainly hemolytic diseases such as sickle cell disease, β-thalassemia, and hemorrhagic conditions, and it has been considered for neurological and parasitic infections such as severe malaria. Our findings may be of interest for therapeutic use as hemopexin peptides can specifically bind and scavenge heme without potential side effects. Hemolysis monitoring is crucial for sepsis and hemolytic diseases. Thus, there is the potential that a diagnostic test can be developed using these sequences.

## Figures and Tables

**Figure 1 ijms-23-03789-f001:**
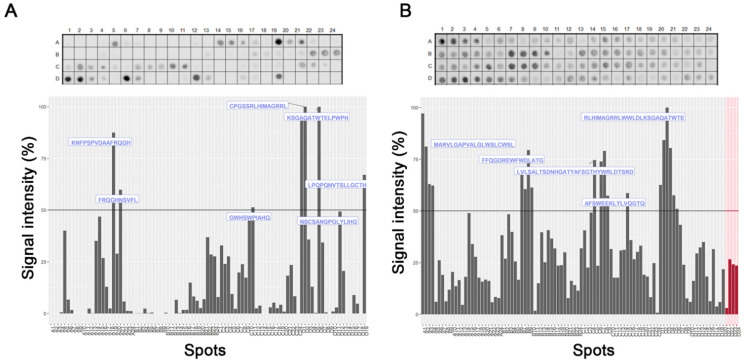
Linear mapping of hemopexin interaction with hemin and hemoglobin in the SPOT synthesis array (15 residues, 10 overlaps). (**A**) The signal intensity of interaction regions of hemopexin sequences with hemin; (**B**) The signal intensity of hemoglobin binding to hemopexin. The negative controls (red bars) represent sequences of 60 kDa chaperonin from *Leptospira interrogans* (DREKLQERLAKLAG), *Toxoplasma gondii*—dense granule protein GRA6 (HPGSVNEFDF), rabies virus (AVNFPNPPGKGGG), and cowpox virus (QEVRKYFCV). Each peptide was identified by spot-synthesis membrane position numbering (Appendix A). Spot intensities below 50% were considered as background.

**Figure 2 ijms-23-03789-f002:**
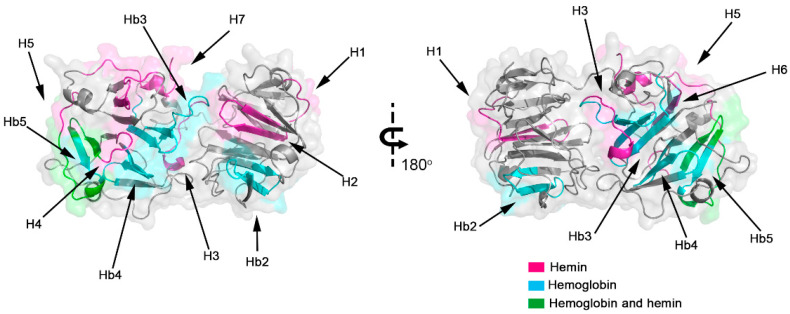
Model of human hemopexin with interaction domains with hemin and hemoglobin. The 3D protein conformation and structure were determined using a homology model, and the interaction regions are highlighted for Hx interaction sites with hemin (magenta), hemoglobin (cyan), and both molecules (green), indicating the positions of the 12 linear sequences identified by spot-synthesis (Table 1).

**Figure 3 ijms-23-03789-f003:**
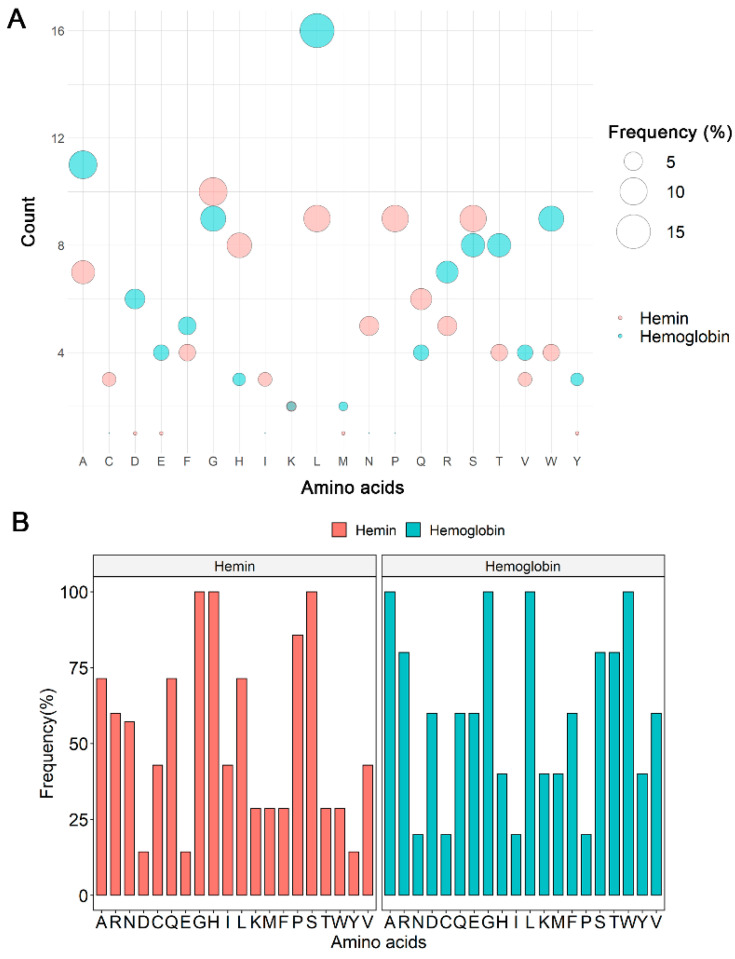
Frequency analysis of the amino acid compositions at Hx interaction sites with hemoglobin and hemin. (**A**) Global counts and frequencies of amino acids in identified Hx sequences. (**B**) Frequencies of amino acids at Hx interaction sites (H1–H7 and Hb1–Hb5).

**Figure 4 ijms-23-03789-f004:**
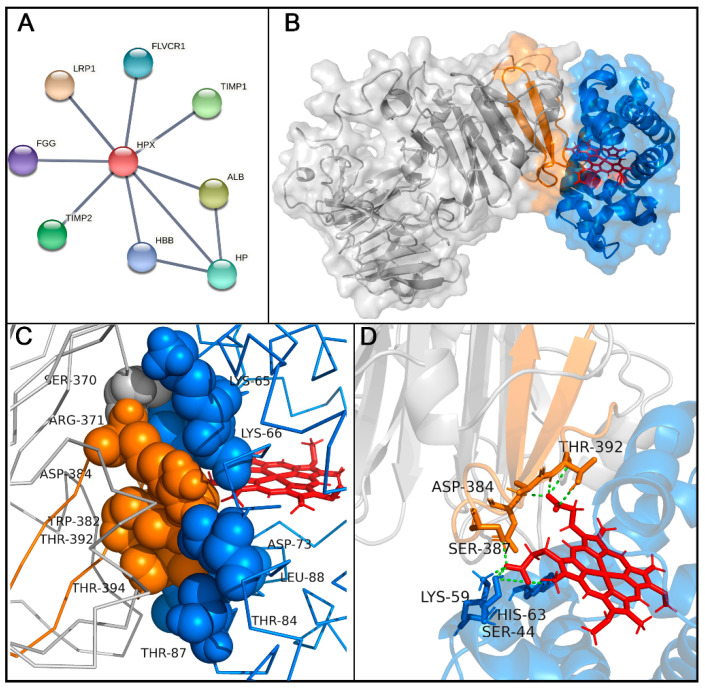
Hemopexin interaction with hemoglobin (Hb) β-chain. (**A**) Network analysis of hemopexin interactome, with Hx interacting with albumin (ALB), haptoglobin (HP), the β-chain of hemoglobin (HBB), the fibrinogen gamma chain (FGG), metalloproteinase inhibitors 1 and 2 (TIMP1 and TIMP2), low-density lipoprotein receptor-related protein 1 (LRP1), and feline leukemia virus subgroup C receptor-related protein 1 (FLVCR1). (**B**) Molecular docking of hemopexin (gray) and active sequence (Hb5; orange) with human hemoglobin β-chain (blue; PDB 2m6z). (**C**) Interaction of hotspot residues between hemoglobin β-chain (blue spheres) and Hx (orange spheres). (**D**) Hydrogen bonds formed by residues in hemoglobin β-chain that interact with heme group (blue sticks and green dashes) and residues in Hx that mediate interaction with heme (orange sticks and green dashes).

**Table 1 ijms-23-03789-t001:** List of the identified interactions between hemopexin and hemin/hemoglobin and the structural properties of these regions.

Code	Interaction Site	Molecule	Position	Secondary Structure
H1	KNFPSPVDAAFRQGH	Hemin	91–105	C + S
H2	FRQGHNSVFL	Hemin	101–110	S
H3	GWHSWPIAHQ	Hemin	291–300	C
H4	CPGSSRLHIMAGRRL	Hemin	366–380	C + S
H5	KSGAQATWTELPWPH	Hemin	386–400	C
H6	NSCSANGPGLYLIHG	Hemin	416–430	C + S
H7	LPQPQNVTSLLGCTH	Hemin	448–462	C
Hb1	MARVLGAPVALGLWSLCWSL	Hemoglobin	1–20	H
Hb2	FFQGDREWFWDLATG	Hemoglobin	161–175	C
Hb3	LVLSALTSDNHGATYAFSGTHYWRLDTSRD	Hemoglobin	261–290	C
Hb4	AFSWEEKLYLVQGTQ	Hemoglobin	311–324	C
Hb5	RLHIMAGRRLWWLDLKSGAQATWTE	Hemoglobin	371–395	C

**Table 2 ijms-23-03789-t002:** Analysis of protein-protein interaction (hydrogen bond) at the hemopexin (Hx) and hemoglobin β-chain (Hb) interface. Hydrogen bond interactions were calculated using the Ligplot algorithm.

Atom	Donor	Residue	Chain	Atom	Acceptor	Residue	Chain	Distance (Å)
OG1	THR	87	B-Hb	O	TRP	393	A-Hx	2.986
NZ	LYS	95	B-Hb	O	ALA	391	A-Hx	2.767
NZ	LYS	95	B-Hb	O	ALA	389	A-Hx	2.897
NZ	LYS	59	B-Hb	O	LYS	386	A-Hx	2.781
NE1	TRP	382	A-Hx	OD2	ASP	73	B-Hb	2.885
NH2	ARG	371	A-Hx	OD1	ASP	73	B-Hb	2.643
NH2	ARG	371	A-Hx	O	GLY	69	B-Hb	3.095
NH1	ARG	371	A-Hx	O	GLY	69	B-Hb	3.008
NZ	LYS	66	B-Hb	OG	SER	370	A-Hx	2.785
NZ	LYS	65	B-Hb	OG	SER	369	A-Hx	2.850
NZ	LYS	65	B-Hb	O	GLY	368	A-Hx	2.914

## Data Availability

The data presented in this study are available in the Appendix A and on request from the corresponding author.

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
