# Peer review of "New Insights into Hemopexin-Binding to Hemin and Hemoglobin"

_ijms, 2022, doi:10.3390/ijms23073789_

Round 1

Reviewer 1 Report

In this study, the authors probe the interaction of hemopexin (Hx) with hemin and hemoglobin employing both experimental and in-silico approaches. The interactions identified can be of interest to a broad  readership with applications in disease monitoring and therapies like in cell injury in malaria, sickle cell disease (SCD), thalassemia, hemorrhage, and hemorrhagic stroke conditions. Thus, the study is quite interesting. However there are some points to be addressed prior to the recommendation for publication:

(1) Right from the beginning, in the abstract, the authors state that hemopexin binds labile Heme with high affinity and crystallographic structures are available. Given the Hx scavenging of Heme with high affinity, the presence of labile Heme could influence the Hx-hemoglobin binding. The authors should discuss this, or present some results on the binding of Hb to Hx in the presence of labile Heme. Allostericity could also play a role, or the relative binding affinity.

(2) What is the "Hp" abbreviation in line 44 of the first page? In general, the first paragraph is quite confusing regarding the interacting partners in protein-protein, protein-Heme pairs in several disorders. Please explicitly present the Hx, Hb, Haptoglobin, Heme interactions known from the literature.

(3) The terms "labile-Heme", "Hemin" used in the text are somewhat confusing, while they are referring to practically the same species.

(4) The authors have used a docking protocol to probe the Hx-Hb interaction. Have thay used the same docking protocol for the Hx-Heme (Hemin) interaction? Do the results compare with the literature? This can be a benchmark of the docking protocol.

(5) Docking protocols are usually for rigid molecules. There are details missing from the docking protocol employed in this study. Moreover, an even short molecular dynamics study on the predicted docked pairs would further elucidate and refine the interactions. For example, the authors state that "works suggest binding heme axially between H105 and H150 [10, 13], but our data demonstrated that only H105 interacted with hemin". The results could be different if dynamics were involved in the prediction (like from Molecular Dynamics simulations).

Overall, I cannot recommend publication of the manuscript in the current form.

Author Response

In this study, the authors probe the interaction of hemopexin (Hx) with hemin and hemoglobin, employing experimental and in-silico approaches. The interactions identified can interest a broad readership with applications in disease monitoring and therapies like in cell injury in malaria, sickle cell disease (SCD), thalassemia, hemorrhage, and hemorrhagic stroke conditions. Thus, the study is quite interesting. However, there are some points to be addressed before the recommendation for publication:

Right from the beginning, in the abstract, the authors state that hemopexin binds labile Heme with high affinity and crystallographic structures are available. Given the Hx scavenging of Heme with high affinity, the presence of labile Heme could influence the Hx-hemoglobin binding. The authors should discuss this or present some results on the binding of Hb to Hx in the company of labile Heme. Allostericity could also play a role or the relative binding affinity.

R: Correct, binding affinity plays a role in the equilibrium. Hrkar, et al., 1973, described the hemopexin extraction of heme from hemoglobin. They stated that hemoglobin is in dynamic equilibria with its constituent parts, subunits, and prosthetic groups. In these equilibria, the fully heme-saturated compounds are favored so that the concentration of free heme in hemoglobin solutions is shallow. Adding a compound to the ferrihemoglobin solution, which possesses a high affinity to heme, causes the shift in chemical equilibrium with the resulting displacement of a portion of heme groups from hemoglobin to the competitor. When free heme is available, hemopexin rapidly binds with a very high affinity (Kd <10−13 M), but free heme concentration is very low in average conditions. Thus, unbound Hx could interact with hemoglobin. In hemolytic conditions, hemoglobin concentration increase rapidly, followed by its degradation into heme. Hx may have a role in the initial scavenging of heme from Hb, providing defense against the toxicity of extracellular hemoglobin.

Additional information was added to the discussion.

(2) What is the "Hp" abbreviation in line 44 of the first page? The first paragraph is quite confusing regarding the interacting partners in protein-protein, protein-Heme pairs in several disorders. Please explicitly present the Hx, Hb, Haptoglobin, Heme interactions known from the literature.

R: We appreciate the suggestion. The "Hp" is the abbreviation of Haptoglobin. We have rewritten the sentence. "However, the exhaustion of Hx and haptoglobin (Hp) binding capacity can lead to oxidative damage." Also, we reconstructed this paragraph to present Hx, Hb, and Hp.

(3) The terms "labile-Heme" "Hemin" used in the text are somewhat confusing, while they are referring to practically the same species.

R: The reference is correct. This nomenclature could lead to misinterpretation, and it was uniformized. Hemin was only maintained in methodology since it was used as a heme source for Spot-synthesis analysis.  

(4) The authors have used a docking protocol to probe the Hx-Hb interaction. Have they used the same docking protocol for the Hx-Heme (Hemin) interaction? Do the results compare with the literature? This can be a benchmark of the docking protocol.

R: We did not perform Hx-heme docking analysis. It was performed previously by another group [Detzel, M.S. et al., Revisiting the interaction of heme with hemopexin. Biol Chem. 2021; 402: 675-691. doi:10.1515/hsz-20 20-0347]

The docking protocol was performed only to analyze interactions of Hb and Hx, aiming at the heme group in Hb and its interactions with identified sequences. The local docking search using RosettaDock server protein-protein interaction could not be used as a benchmark since it used a different algorithm for protein-protein interaction and protein-ligand interactions.  

(5) Docking protocols are usually for rigid molecules. Details are missing from the docking protocol employed in this study. Moreover, an even short molecular dynamics study on the predicted docked pairs would further elucidate and refine the interactions. For example, the authors state that "works suggest binding heme axially between H105 and H150 [10, 13], but our data demonstrated that only H105 interacted with hemin". The results could be different if dynamics were involved in the prediction (like from Molecular Dynamics simulations).

R: The RosettaDock Server performs a local rigid docking search. The algorithm searches a set of conformations near the given starting conformation for the optimal fit between the two partners. Previously analysis was directed to the sequence in Hx with the highest spot signal (Hb5). The initial position of partners was performed using experimental data from spot-synthesis. The statement refers to empirical data; H150 present in peptides had low signal intensity. We understand that molecular dynamics could complement experimental data. A more robust in silico analysis including molecular dynamics is a future perspective, along with the synthesis of identified peptides and measurement of binding affinity. As suggested, we performed further research to refine the docking data. We performed a different approach using HADDOCK (High Ambiguity Driven protein-protein DOCKing), an information-driven flexible docking server, to model biomolecular complexes. The HADDOCK 2.4 platform for protein-protein interactions analysis is guided by the prediction of likely residues (called ambiguous interaction restraints (AIRs)) found at the interface. These residues may be active (interacting residue) or passive (nearby interacting residue). Hemopexin active residues were defined as Hb5 (RLHIMAGRRL WWLDLKSGAQATWTE) retrieved from Spot-synthesis data. The residues surrounding the heme group in Hemoglobin were selected as passive. Several interaction poses were clustered, creating 126 structures in 13 clusters, which represents 63 % of the water-refined models generated. HADDOCK webserver ranks structures by biochemical and/or biophysical in clusters. Cluster 8 was considered the most reliable based on obtained scores, the structures in cluster 8 were carefully analyzed, and the lowest energy structure (-102 Kcal/mol) was chosen for further analysis. Additional information can be visualized in the supplemental material. Also, hotspot amino acid residues prediction and hydrogen bonds were analyzed. 

Reviewer 2 Report

This manuscript is very informative and will be of great help to researchers in the field. Experiments are sounds and well presented. I recommend revision of the Introduction and Discussion session: some sentences are not grammatically clear, and the overall impression is a lack of coherence. 

Author Response

This manuscript is very informative and will significantly help researchers in the field. Experiments are sounds and well presented. However, I recommend revising the Introduction and Discussion session: some sentences are not grammatically clear, and the overall impression lacks coherence.

R: We appreciate the referee's comments. We addressed new information in Introduction and Discussion. Also, rewrote the sentences to improve the coherence and cohesion of the manuscript.  

Round 2

Reviewer 1 Report

The authos have adequately ammended the manuscript, as per reviwer suggestions.